# Independent Reproduction of the FLASH Effect on the Gastrointestinal Tract: A Multi-Institutional Comparative Study

**DOI:** 10.3390/cancers15072121

**Published:** 2023-04-02

**Authors:** Anet Valdés Zayas, Neeraj Kumari, Kevin Liu, Denae Neill, Abagail Delahoussaye, Patrik Gonçalves Jorge, Reiner Geyer, Steven H. Lin, Claude Bailat, François Bochud, Raphael Moeckli, Albert C. Koong, Jean Bourhis, Cullen M. Taniguchi, Fernanda G. Herrera, Emil Schüler

**Affiliations:** 1Radio-Oncology Department, AGORA Cancer Research Institute, Lausanne University Hospital, Lausanne University, Rue du Bugnon 46, CH-1011 Lausanne, Switzerland; 2Department of Radiation Oncology, The University of Texas MD Anderson Cancer Center, Houston, TX 77030, USA; 3Department of Radiation Physics, The University of Texas MD Anderson Cancer Center, Houston, TX 77030, USA; 4Graduate School of Biomedical Sciences, The University of Texas, Houston, TX 77030, USA; 5Institute of Radiation Physics, Lausanne University Hospital, Lausanne University, Rue du Grand-Pré-1, CH-1007 Lausanne, Switzerland

**Keywords:** ultra-high dose rate, FLASH, radiation oncology, radiation therapy, multi-institutional, gastrointestinal toxicity, radiation response, crypt assay

## Abstract

**Simple Summary:**

The ability of ultra-high dose rate FLASH radiation therapy (RT) to reduce normal tissue toxicity without affecting tumor response relative to conventional dose rate radiation therapy could fundamentally change the way we treat cancer. However, this field is still in its early stages, and the magnitude of the sparing effect between treatment centers differs greatly for reasons as yet unknown, which has put the robustness of the effect into question. In this study, we show that when similar irradiation beam parameter settings are used, the induced sparing effect is robust and reproducible across institutions. These settings should serve as a reference for further optimization of the FLASH effect.

**Abstract:**

FLASH radiation therapy (RT) is a promising new paradigm in radiation oncology. However, a major question that remains is the robustness and reproducibility of the FLASH effect when different irradiators are used on animals or patients with different genetic backgrounds, diets, and microbiomes, all of which can influence the effects of radiation on normal tissues. To address questions of rigor and reproducibility across different centers, we analyzed independent data sets from The University of Texas MD Anderson Cancer Center and from Lausanne University (CHUV). Both centers investigated acute effects after total abdominal irradiation to C57BL/6 animals delivered by the FLASH Mobetron system. The two centers used similar beam parameters but otherwise conducted the studies independently. The FLASH-enabled animal survival and intestinal crypt regeneration after irradiation were comparable between the two centers. These findings, together with previously published data using a converted linear accelerator, show that a robust and reproducible FLASH effect can be induced as long as the same set of irradiation parameters are used.

## 1. Introduction

Ultra-high dose rate FLASH radiation therapy (RT) has recently emerged as a promising new method that has substantially different effects on tumors and normal tissues, with substantially reduced treatment-related toxicity, relative to conventional dose rate (CONV) RT. In FLASH RT, radiation is given at ultra-rapid dose rates that are thousands of times higher than those currently used in CONV RT in clinical practice. Although FLASH RT and CONV RT were initially distinguished by the mean dose rates of the radiation beam (≥40 Gy/s for FLASH RT vs. 0.01 Gy/s for CONV RT), the full definition is more complex and likely includes several interdependent variables, such as the number and width of radiation pulses, the pulse repetition rate, and the total duration of exposure (<200 ms for FLASH RT vs. several minutes for CONV RT) [1,2,3].

Increasing evidence suggests that FLASH RT spares normal tissues to a greater extent than CONV RT while maintaining iso-effectiveness in terms of tumor control [1,3,4]. This so-called “FLASH effect” has been demonstrated in a variety of animal models and organ systems [5,6,7,8]. The clinical translation of FLASH RT was also recently demonstrated in a patient with recurrent T cell lymphoma, for which FLASH RT was deemed feasible, safe, and effective [9,10]. The first Phase 1 clinical trial using FLASH RT was also recently concluded on patients with painful bone metastases to the extremities. Using a shoot-through scanning proton beam, the team at Cincinnati Children’s Hospital was again able to show the feasibility and safety of this new modality [11,12].

Although FLASH RT has shown promising effects in preclinical and clinical studies, the question remains of the robustness and reproducibility of the FLASH effect across centers and across irradiators, especially when used on animals or patients with different genetic backgrounds, diets, and microbiomes, all of which can influence the effects of radiation on normal tissue [13,14,15]. The magnitude of the FLASH effect varies widely at different treatment centers for reasons that are currently unknown, but these differences may well be related to differences in the irradiation parameters and biological endpoints used. One fundamental requirement seems to be a short delivery time (<200 ms) [5], and another is a minimum dose rate of 40 Gy/s. The FLASH effect also seems to require a total dose of at least 10 Gy and may increase as a function of the dose, depending on the organ being irradiated [16].

To address questions of rigor and reproducibility across different centers, we analyzed independent data sets from Lausanne University (CHUV) in Vaud, Switzerland and from MD Anderson Cancer Center in Houston, TX, USA. Investigators at the two institutions devised study designs and data acquisition processes separately and used similar radiation parameters and setups to investigate the FLASH effect in mice after total abdominal irradiation. We demonstrated that by using a set of beam parameters known to induce the FLASH effect, FLASH effects of similar magnitude were successfully induced at both institutions, indicating the robustness of the FLASH effect across institutions.

## 2. Materials and Methods

### 2.1. MD Anderson

#### 2.1.1. Animals

Eight-week-old female C57BL/6 mice were purchased from Jackson Laboratories (Bar Harbor, ME, USA). The mice had access to standard feed and sterile water ad libitum and were maintained on a 12-h light/dark cycle. All mouse procedures used were approved by the Institutional Animal Care and Use Committee (IACUC) of The University of Texas MD Anderson Cancer Center.

#### 2.1.2. Irradiation Setup and Parameters

All irradiations were given with a FLASH Mobetron unit (IntraOp Medical, Sunnyvale, CA, USA) [17,18]. A custom-made jig was used to immobilize the mice, after which the mice were sedated with isoflurane and air as carrier gas. A customized collimator with the mouse jig indexed to this collimator was used to allow reproducible setups for total abdominal irradiation. For the survival analysis, mice (n = 10/group) were irradiated with a single dose of either 15.5 Gy or 17 Gy; for the intestinal crypt assay, mice (n = 5–10/group) were irradiated with a single dose of 11, 12, 13, or 14 Gy. The mouse irradiation setup was calibrated using EBT3 film (Ashland, Bridgewater, NJ, USA), as previously described [19]. CONV dose rate irradiations were performed with 9-MeV electrons also delivered with the Mobetron at a dose rate of 0.17 Gy/s (7 mGy/pulse). The FLASH beam parameters are shown in Table 1.

#### 2.1.3. Endpoints

Signs of acute toxicity (e.g., weight loss, sepsis, hunching, diarrhea, activity, behavioral changes) were monitored throughout the 30-day experiment, and gastrointestinal (GI) toxicity was assessed with a crypt regeneration assay and survival analysis [8,20,21,22]. For the crypt assay, mice were euthanized with CO_2_ inhalation followed by cervical dislocation. The jejunum from each mouse was collected at 84 h (3.5 days) after irradiation, fixed in 10% neutral buffered formalin for 24 h, washed once with phosphate-buffered saline (PBS), and stored in 70% ethanol. All tissues were embedded in paraffin, and nine 3-μm-thick transverse sections were collected and stained with hematoxylin and eosin (H&E). Regenerated crypts were counted manually based on the presence of (1) U-shaped structure with a lumen, (2) structure along the circumferential edge, (3) clear, basophilic structure, and (4) multicellularity (at least 10 cells).

For the survival analysis, mice were weighed at the time of irradiation and then every other day thereafter until day 30. Mice were euthanized early (i.e., before the end of the 30-day experimental period) if the following IACUC-approved criteria were met: (1) becoming moribund (exhibiting hunched posture, non-weight-bearing lameness, ruffled fur, labored breathing) or (2) losing more than 20% of the baseline body weight.

#### 2.1.4. Statistical Analysis

Mouse survival was analyzed using the Kaplan–Meier method, and curves were compared with log-rank tests. Grouped data sets were analyzed with two-way analysis of variance. Data are shown as means ± SEM. All statistical analyses were performed with GraphPad Prism V.8 (La Jolla, CA, USA), and *p*-values of <0.05 were considered to indicate significant differences.

### 2.2. CHUV

#### 2.2.1. Animals

Eight- to ten-week-old male C57BL/6 mice were purchased from Jackson Laboratories and were provided water and food (Scientific Animal Food & Engineering, R150) ad libitum and maintained on a 12-h light/dark cycle at 22 °C ± 2 °C with a relative humidity of 55% ± 10%. Mice were anesthetized before irradiation with an intraperitoneal injection of ketamine (75 mg/kg) with medetomidine (0.5 mg/kg). All animal experiments were approved by the Animal Ethics Committee of Vaud, Switzerland (VD3677b) and performed according to institutional guidelines.

#### 2.2.2. Irradiation Setup and Parameters

All mice were irradiated with a FLASH Mobetron device (IntraOp Medical, Sunnyvale, CA, USA), and the same machine was used to deliver radiation at FLASH and CONV dose rates. The commissioning of the machine and its characteristics were extensively described elsewhere [17]. A round 4 cm diameter collimator was used together with a cylindrical applicator of 20 cm length for whole abdominal irradiations. Mice were held by two 1.5 cm thick solid water blocks on their sides during irradiations to ensure reproducible positioning. They received a single fraction of 16 Gy for both modalities of radiation. FLASH beam parameters are shown in Table 1. For CONV irradiations, mice were irradiated with the 9 MeV electron beam with a mean dose rate of 0.17 Gy/s corresponding to a dose per pulse of approximately 7 mGy. Dosimetric preparation was performed with alanine pellets from Harwell Dosimeters, as described in a previous study [23].

#### 2.2.3. Endpoints

For the survival crypt count assays, the mice were euthanized by CO_2_ asphyxiation followed by cervical dislocation at 24, 48, or 72 h after irradiation, and the mid-segment of the small intestine corresponding to the jejunum was harvested. Paraffin sections were processed as described elsewhere [24]. Briefly, the jejunum was cleaned, cut open longitudinally, pinned, and fixed in paraformaldehyde for 24 h, washed once with PBS, “Swiss-rolled,” and then stored in 70% ethanol. The fixed tissues were then processed, embedded in paraffin, sliced in 5-μm sections, and stained with H&E. Surviving crypts were counted manually based on the following criteria: (1) mucosa oriented perpendicular to the long axis of the intestine and (2) the presence of crypts consisted of ≥5 basophilic crypt epithelial cells. Slides were analyzed digitally with QuPath version 0.3.0. Crypts were counted in four randomly selected, identically sized areas of the Swiss-rolled specimens per mouse, and results are reported as the average number of surviving crypts per µm^2^.

Survival time was measured from irradiation until euthanasia. Signs of gastrointestinal injury (e.g., body weight loss, diarrhea, and decreased activity) were documented, scored, and used to determine if euthanasia was required as follows: body weight loss of >15%; or a combination of body weight loss of ≤15% + immobility; or a combination of body weight loss of ≤15% + diarrhea; or a combination of body weight loss + immobility + diarrhea). All mice surviving until 30 days were euthanized at that time.

#### 2.2.4. Statistical Analysis

Survival was analyzed using the Kaplan–Meier method, and curves were compared with log-rank tests. Two-tailed unpaired Student’s *t*-tests were used to determine significant differences between two means. Comparisons involving three or more groups were performed with analysis of variance, and pairwise comparisons in post hoc analyses were performed with a Tukey adjustment. Data are shown as mean ± SEM. All statistical analyses were performed with GraphPad Prism V.9 (CHUV), and *p*-values of <0.05 were considered to indicate significant differences.

## 3. Results

Both institutions investigated the acute effects of whole abdominal irradiation delivered as FLASH RT or CONV RT for up to 30 days after treatment. Both institutions monitored mice for signs of morbidity (weight loss, sepsis, hunching, diarrhea, activity, behavioral changes) and assessed overall survival at 30 days. Jejunum specimens were harvested from subsets of mice at 84 h after irradiation in the MD Anderson experiments and at 24, 48, and 72 h after irradiation in the CHUV experiments.

### 3.1. MD Anderson Experiments

The study schedule for the experiments conducted at MD Anderson is illustrated schematically in Figure 1A.

#### 3.1.1. Body Weight

Reductions in body weight were noted in mice after FLASH or CONV RT at both centers. At MD Anderson, the maximum reduction in body weight was reached at 8 days after irradiation with either FLASH RT or CONV RT (data not shown). Some of the mice (those that survived until day 30) regained body weight after day 8, and others (those that died or were euthanized before 30 days) continued to lose weight.

#### 3.1.2. Survival

The survival rates at 30 days after a 15.5-Gy dose were 100% in the FLASH RT group and 90% in the CONV RT group. Survival rates after a 17-Gy dose were 40% after FLASH RT and 0% after CONV RT (Figure 1B).

#### 3.1.3. Histologic Evaluation and Crypt Assays

The pathophysiology of gastrointestinal injury after irradiation is known to result from malfunctioning clonogenic compartments at the base of the Lieberkühn crypts [25,26]. We sought to evaluate if the improved outcome of FLASH RT in lowering gastrointestinal toxicity was due to better preservation of the small intestine mucosal architecture.

In the MD Anderson experiments, histologic evaluations at 84 h (3.5 days) after irradiation (Figure 1C) revealed significant reductions in the numbers of regenerating crypts per circumference, with the extent of the reduction increasing with dose in both the CONV RT and FLASH RT groups (Figure 1D). However, the numbers of regenerated crypts were higher after FLASH RT than after CONV RT. Slightly lower doses (11–14 Gy) were used for the regenerating crypt assay to allow sufficient data for statistical analysis. The maximum separation in response between FLASH RT and CONV RT was observed at 13 Gy, where the number of regenerating crypts was >300% higher after FLASH RT than after CONV RT (Figure 1D).

### 3.2. CHUV Experiments

The study schedule for the study conducted at CHUV is illustrated schematically in Figure 2A.

#### 3.2.1. Body Weight

At CHUV, mice in both the FLASH RT and CONV RT groups began to lose weight on day 3 after irradiation, with the maximum weight loss on day 5. Body weight began to recover starting on day 10, and all surviving mice had regained their original body weight by day 22 (data not shown).

#### 3.2.2. Survival

A single 16-Gy dose was used for both survival and normal tissue toxicity evaluation. The survival rate at 30 days post-irradiation was 60% in the FLASH RT group and 45% in the CONV RT group (Figure 2B).

#### 3.2.3. Histologic Evaluation and Crypt Assays

Histologic analysis of jejunum specimens (Figure 2C) revealed no discernible harm to the intestinal epithelia at 24 h (Figure 2D), but all irradiated mice showed radiation-induced mucosal damage (measured by the number of surviving crypts) at 48 h and 72 h. Similar to the MD Anderson experiments on regenerating crypts, mice in the FLASH RT group had higher numbers of surviving crypts than did the CONV RT group (Figure 2D).

## 4. Discussion

The considerable variability in the magnitude of the FLASH effect across the published literature has raised questions regarding the robustness and the reproducibility of that effect. We believe that this variability probably results from inconsistencies between studies in the choice of endpoints and beam parameters. However, the published literature suggests that induction of a FLASH effect involves more than simply delivering radiation at dose rates in excess of 40 Gy/s.

Despite minor differences in study design between MD Anderson and CHUV, our findings nevertheless indicate that the use of similar radiation beam parameters can result in FLASH effects of similar magnitude. However, the findings between the two sites differed somewhat because of differences in the endpoints studied, thereby making direct comparisons difficult. The differences in survival data resulted from the use of different criteria for euthanasia at MD Anderson vs. CHUV; those criteria reflect differences in site-specific animal care regulations (i.e., the maximum body weight loss allowed without euthanasia). Another difference was that the MD Anderson experiments used female mice and the CHUV experiments used male mice. Others have shown that irradiation affects male and female mice differently [27,28]. Other potential inter-institutional differences may have included the average age and weight of mice at the time of irradiation as well as procedures for animal handling and maintenance. Another important difference factor is the timing and nature of the intestinal crypt assays. The MD Anderson group counted the number of regenerating crypts at 84 h using the classical protocol from Withers and Elkind [22]. The CHUV group counted the number of surviving intestinal crypts within 72 h after irradiation, which is within the window of turnover of the murine small intestines, which is 3–4 days [29]. Despite these minor methodological differences, the results from two institutions using the same irradiators and the same stock of mice found similar levels of FLASH effect in the GI tract.

Regardless of these differences, however, our findings are also comparable with previously reported studies of GI toxicity after electron irradiation [8]. Although that study used slightly different beam parameters (a 16-MeV electron beam, 108 Hz, 2 Gy/pulse, 216 Gy/s, and 4E5 Gy/s, i.e., instantaneous dose rate), the numbers of regenerating crypts and survival rates were similar to what was found in the current study.

A key take-home message from this work is that using beam parameters that are known to evoke FLASH effects can lead to responses that are robust and reproducible across machines and institutions. However, to determine the true boundaries that must be worked within to induce the FLASH effect reproducibly, more comprehensive investigations are needed, in which each individual beam parameter is varied independently of the others [3]. Bourhis et al. made the first attempt to elucidate the requirements for inducing the FLASH effect in 2019 and again in 2021 [5,30]. These studies have indicated that overall irradiation time and dose rate within each pulse delivered are critical to the FLASH effect but did not include evaluations of dose per pulse, pulse width, or total dose delivered. Another recent analysis demonstrated that the FLASH effect may indeed depend on the dose delivered. When reviewing the published literature on comparing CONV vs. FLASH RT following whole abdominal irradiation, doses of up to 10 Gy resulted in minimal normal-tissue sparing, while doses higher than 10 Gy exhibited a concordantly higher FLASH effect, with a maximal sparing at 14–16 Gy [31]. That finding is consistent with the trends shown in the current study, in which sparing was greatest at doses of 13–14 Gy, but lower and higher doses had a lesser sparing effect (data not shown). That said, the offset in dose could be explained by the location where the dose is defined. In the current study, we reported the radiation dose to the center of the mouse, which is approximately 5% lower than the surface dose reported in the comparative data set [8,31].

## 5. Conclusions

This is the first study to demonstrate the physical and biological effects of FLASH RT achieved with a FLASH Mobetron unit. Two independent institutions were able to demonstrate similar biological sparing effects resulting from high-dose, single-fraction abdominal irradiation delivered with FLASH RT versus CONV RT. The FLASH Mobetron was available to both institutions, and the beam parameters used were nearly identical. The most striking finding was that, despite differences in selected endpoints and in euthanasia criteria, both institutions found that FLASH RT produced significantly different responses than CONV RT. When comparing to and including previously published literature using similar endpoints and with similar irradiation beam parameters but on different irradiation units, we can show that the FLASH effect is robust and reproducible regardless of the unit used, as long as the same set of beam parameters are used.

## Figures and Tables

**Figure 1 cancers-15-02121-f001:**
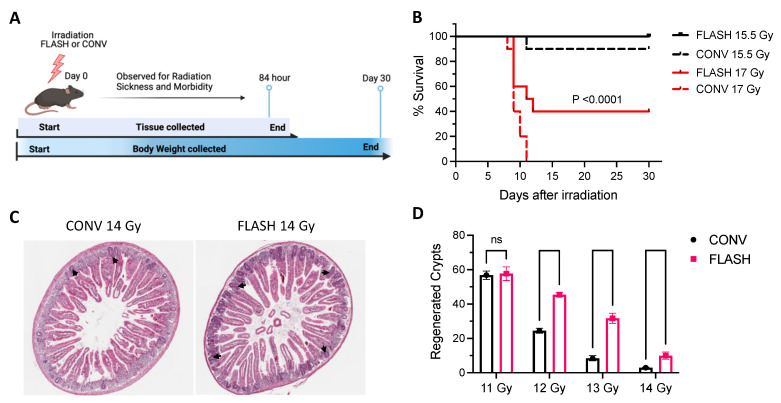
At MD Anderson, ultra-high dose rate FLASH radiation therapy (RT) led to higher survival and reduced gastrointestinal toxicity compared with conventional dose rate (CONV) RT in mice subjected to whole-abdominal irradiation. (**A**) Schematic of radiation schedule treatment and follow-up at MD Anderson. (**B**) Kaplan–Meier plot shows percent survival at 30 days after irradiation with FLASH RT or CONV RT at two different doses (n = 10/group). Survival rates were higher for the mice treated with 15.5 Gy than for mice treated with 17 Gy in both groups (*p* < 0.0001, log-rank test). FLASH-treated mice showed higher survival than CONV-treated mice. (**C**) H&E-stained transverse sections of jejunum at 84 h after a single 14-Gy dose showing differential response in regenerated crypts. (**D**) Numbers of regenerated crypts per circumference for mice given 11, 12, 13, or 14 Gy by CONV RT or by FLASH RT. Number of regenerated crypts are significantly higher in all FLASH groups compared to the dose-matched CONV groups at doses of 12 Gy and higher. ns–not significant, unpaired Student’s *t*-tests.

**Figure 2 cancers-15-02121-f002:**
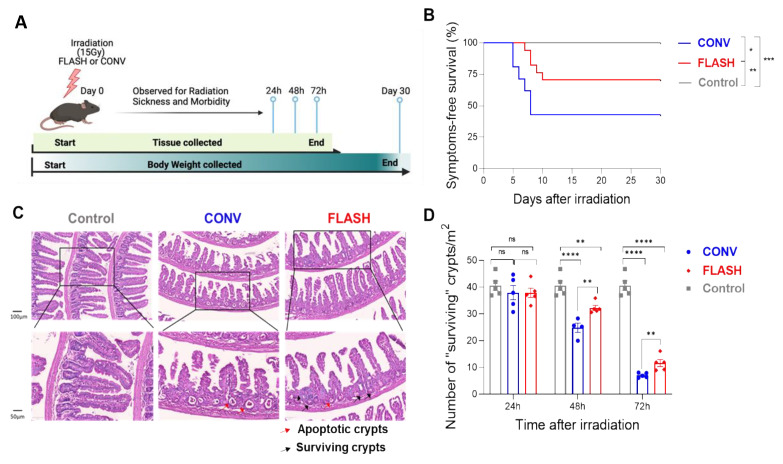
At CHUV, ultra-high dose rate FLASH RT led to higher survival and reduced gastrointestinal toxicity compared with CONV dose rate RT. (**A**) Schematic for radiation treatment and follow-up studies at CHUV. (**B**) Kaplan–Meier plot of survival up to 30 days post-irradiation (mice were euthanized after meeting the CHUV criteria for morbidity.). Data shown are mean ± SEM pooled from 2 independent experiments (21 mice in the CONV RT group, 17 mice in FLASH RT, and 20 mice in Control). FLASH RT-treated mice survived longer than CONV RT-treated mice. (**C**) H&E-stained Swiss-roll sections of jejunum; scale bar in upper panels indicates 100 µm; scale bar in the magnified view in the lower panels is 50 µm. Red arrows point to apoptotic crypts and black arrows to surviving crypts. (**D**) Numbers of surviving crypts at 24–72 h after a single 16-Gy dose of CONV or FLASH RT. The numbers of surviving crypts were no different between the Control, CONV, and FLASH groups at 24 h, but were significantly higher in the FLASH group vs. the CONV group at 48 and 72 h. ns–not significant, * *p*< 0.05; ** *p* < 0.001; *** *p* < 0.0001; **** *p* < 0.00001 (log-rank test).

**Table 1 cancers-15-02121-t001:** Radiation beam parameters used at both institutions.

Parameters	Values at MD Anderson	Values at CHUV
**Nominal electron beam energy**	9 MeV	9 MeV
**Total absorbed dose ***	10.5–17 Gy	15.8–16.3 Gy
**Number of pulses**	7–12	8
**Fractionation schedule**	Single fraction	Single fraction
**Mean dose rate ***	185–225 Gy/s	199 Gy/s
**Instantaneous dose rate ***	1.24 × 10^6^ ± 5.74 × 10^4^ Gy/s	0.97 × 10^6^ Gy/s
**Pulse frequency**	120 Hz	90 Hz
**Dose per pulse ***	1.5 ± 0.07 Gy	1.94 Gy
**Pulse width**	1.2 µs	2 µs
**Duration of exposure**	42–75 ms	78 ms
**Beam field size ****	40 × 40 mm	40 mm diameter

* Value at the center of the mouse (8 mm depth); ** Physical dimension of collimation.

## Data Availability

The data presented in this study are available on request.

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
