# Peer review of "Independent Reproduction of the FLASH Effect on the Gastrointestinal Tract: A Multi-Institutional Comparative Study"

_cancers, 2023, doi:10.3390/cancers15072121_

Round 1
Reviewer 1 Report
In the past few years FLASH radiotherapy (FLASH RT) has gained more attention in preclinical studies as the next important improvement in anti-tumoral treatment, overcoming conventional RT (CONV RT). The greatest advantage is that the ultra-fast high dose delivered with FLASH RT can maintain the equal, if not major, toxic effect on cancerous cells and at the same time can show a sparing effect on normal healthy tissue surrounding the tumour site compared to CONV RT. This sparing effect is called the FLASH effect. However, in vitro and in vivo data available on the FLASH effect are not sufficient to transpose the study on humans. Moreover the robustness and reproducibility of the FLASH effect across centers and across irradiators remain an open question, particularly when used on animals with different genetic backgrounds, diets, and microbiomes, all of which can affect the effects of radiation on normal tissue. In this study the authors analyzed separate data sets two Centers to address issues of rigor and reproducibility across several centers. To evaluate the FLASH impact in mice after total abdominal irradiation, researchers at the two institutions independently developed study designs, data gathering procedures, and radiation parameters and setups. The authors showed that the FLASH effect could be successfully created using a set of beam settings known to cause it, demonstrating the FLASH phenomenon's inter-institutional stability. The work is well presented and very interesting, given the growing interest in this topic and the increasing number of in vivo studies performed.
Author Response
Thank you for this positive feedback.
Reviewer 2 Report
The paper aims to compare two radiotherapy methods in two different institutes. The manuscript is clear, relevant to the field, and well-structured. The cited references mostly recent publications and relevant.The paper isscientifically sound and the experimental designs are appropriate for testing and comparing the results.
The figures are clear and easy to understand.
The conclusions are appropeiate to the presented results.
Minor comments to check:
1. In table 1: it appears "1.24x106" - is that correct?
2. Line 286: "108 Hz" - is that correct?
Author Response
Comment 1. In table 1: it appears "1.24x106" - is that correct?
Response: It was a typing error we corrected this. It is 1.24x106
Comment 2:Line 286: "108 Hz" - is that correct?
Response: Yes, it is correct.
Reviewer 3 Report
Paper overview
The main goal of the paper is to show the reproducibility of the FLASH effect by performing similar experiments (beam parameters etc.) at two institutions on normal tissue.
Overall the paper is clear and well written. The of reproducibility of the FLASH effect is also an important topic considering the increasing number of groups interested in FLASH radiotherapy .
The negative aspect of the paper is that the reproducibility evaluation of the FLASH effect was performed considering only on 3 main endpoints related to the outcome of abdominal irradiation. I believe that adding more end points related to other irradiation sites (e.g brain) will make the paper main message more robust.
Author Response
Response: We agree with the reviewer that adding more radiation sites other than the abdomen would absolutely add more impact to the study. However, the focus of our study is limited to intestinal toxicity and any additional site is beyond the current scope. The main message of the current study is the robustness of the FLASH effect in abdominal irradiation. Further studies in a broader range of irradiation sites and institutions are currently in the planning phase.
Round 2
Reviewer 3 Report
No further comments.